# Physiological Femoral Condylar Morphology in Adult Knees—A MRI Study of 517 Patients

**DOI:** 10.3390/diagnostics13030350

**Published:** 2023-01-18

**Authors:** Marc-Pascal Meier, Yara Hochrein, Dominik Saul, Mark-Tilmann Seitz, Paul Jonathan Roch, Katharina Jäckle, Ali Seif Amir Hosseini, Wolfgang Lehmann, Thelonius Hawellek

**Affiliations:** 1Department of Trauma Surgery, Orthopaedics and Plastic Surgery, University Medical Center Goettingen, Robert-Koch-Straße 40, 37075 Göttingen, Germany; 2Department of Trauma and Reconstructive Surgery, BG Trauma Center Tuebingen, Eberhard Karls University Tuebingen, 72076 Tübingen, Germany; 3Division of Endocrinology, Kogod Center on Aging, Mayo Clinic, Rochester, MN 55905, USA; 4Department of Diagnostic and Interventional Radiology, University Medical Center Goettingen, Robert-Koch-Straße 40, 37075 Göttingen, Germany

**Keywords:** condylar offset, condylar morphology, MRI, knee morphology, knee arthroplasty

## Abstract

Background: In the age of individualised arthroplasty, the question arises whether currently available standard implants adequately consider femoral condylar morphology (FCM). Therefore, physiological reference values of FCM are needed. The aim was to establish physiological reference values for anterior (ACO) and posterior condylar offset (PCO) as well as for the length of the medial (LMC) and lateral femoral condyles (LLC). Methods: The knee joints of 517 patients (mean age: 52.3 years (±16.8)) were analysed retrospectively using MRI images. Medial (med) and lateral (lat) ACO and PCO, as well as LMC and LLC, were measured. All FCM parameters were examined for association with age, gender, side and osteoarthritis. Results: Mean ACOmed was 2.8 mm (±2.5), mean ACOlat was 6.7 mm (±2.3), mean PCOmed was 25.7 mm (±4.6), mean PCOlat was 23.6 mm (±3.0), mean LMC was 63.7 mm (±5.0) and mean LLC was 64.4 mm (±5.0). Except for PCOmed, the mean values of all other FCM parameters were significantly higher in male knees compared to female knees. ACOmed and PCOmed showed significant side-specific differences. There were no significant differences in relation to age and osteoarthritis. Conclusion: The study showed significant differences in FCM side- and gender-specifically in adult knees. These aspects should be considered in the discussion of individual and gender-specific knee joint replacement.

## 1. Introduction

The morphology of the adult knee joint is complex [1,2,3,4]. Therefore, an exact reconstruction of the physiological knee morphology by implants for knee joint replacement is difficult [5,6,7,8]. One approach is to use individually designed implants based on computed tomographic images [9,10]. Furthermore, the necessity of gender-specific implants is a discussed topic in the current literature [11,12,13].

In particular, the morphology of the distal femur is difficult to reconstruct [5,7]. Regarding the age of individualised arthroplasty, the question arises whether currently available standard implants adequately represent femoral condylar morphology (FCM) [7,11]. The anterior (ACO) and posterior (PCO) condylar femoral offset have an influence on postoperative knee joint function and patients’ satisfaction after knee arthroplasty [5,14,15,16,17,18]. In this context, Zheng et al. analysed the phenomenon wherein patients describe the feeling of an increase in the size of their own knee after primary implantation of a total knee arthroplasty [5]. The authors called this “patient-perceived enlargement of knee” (PPEK). The authors demonstrated that patients with PPEK have worse postoperative functional scores and satisfaction levels, while radiological analysis showed a significantly higher ACO and smaller PCO postoperatively in PPEK. In addition, increased ACO is associated with the occurrence of anterior knee pain [5,15]. An increase in patellofemoral contact pressure is assumed to be the causative factor [5,19]. There is ambiguity in the literature regarding the influence of PCO on postoperative outcome after total knee arthroplasty [16,17,18,20]. Some studies suggest that a reduction in PCO leads to bony impingement and flexion inhibition [18,20]. Other studies have failed to find associations between an increase or decrease in PCO with postoperative range of motion/occurrence of knee pain [16,17].

It can be concluded that FCM has a significant influence on postoperative outcome and patient satisfaction after total knee arthroplasty. There are findings which indicate that the current standard implants do not sufficiently reconstruct physiological FCM, which can lead to complaints in the form of knee pain and reduced mobility [7,11].

However, to the authors’ knowledge, no study exists in the current literature that provides reference values for physiological FCM based on a radiological examination of patients without manifest osteoarthritis.

Therefore, the aim of the present study was to investigate the physiological FCM retrospectively using magnetic resonance imaging (MRI), and to analyse the presence of side-, age-, gender- and joint degeneration-specific differences.

## 2. Materials and Methods

### 2.1. Patients

Between 2007 and 2020, a total of *n* = 5627 patients underwent magnetic resonance imaging of the knee joint in the Department for Diagnostic and Interventional Radiology of the University Medical Center Goettingen. These MRIs were reviewed retrospectively. After applying inclusion and exclusion criteria, 517 patients were included in the final analysis. The study collective was divided into two age groups (20–50 and >50 years). The study was approved by the local ethics committee (IRB number: 35/7/20) and performed in accordance with the principles expressed in the Declaration of Helsinki. Without exception, the evaluated MRIs were taken as part of routine diagnostics because of clinical symptoms. All MRIs were assessed by a senior radiologist and Y.H., M.-P.M., A.S.A.H. and T.H. to exclude extended structural injuries or heavy joint degeneration.

### 2.2. Inclusion Criteria

All examinations accessible via the PACS system (Picture Archiving and Communication System) between 1 January 2007 and 31 December 2020 were initially included in the study. Out of these, all patients with an age of 20 years or more were included. All MRI scans were performed on patients to assess knee joint pathologies. A Kellgren/Lawrence score [21] ≥3 was classified as manifest osteoarthritis. The study collective was divided up in two groups (without and with radiological osteoarthritis). All MRIs were examined by the internal radiology department as part of the clinical diagnostic procedure. Every report was re-evaluated by Y.H., M.-P.M., A.S.A.H. and T.H. in a blinded fashion.

### 2.3. Exclusion Criteria

All patients with fractures, osteonecrosis, dysplasia, ligament damage or tumours were excluded. Patients who had undergone osteosynthesis or arthroplasty were likewise excluded. Similarly, the data did not include patients who had any other implants after knee joint preservation surgery. Low-quality MRIs (based on only a few gates), were excluded. In addition, all MRIs with imaging artefacts were ruled out.

### 2.4. MRI Analysis, Parameters and Methods of Measurement

All measurements were taken via the PACS system (Picture Archiving and Communication System). Software from GE Healthcare, called Centricity^TM^ Universal Viewer, was used (RA1000, edition 2019, Buckinghamshire, UK). The osteoarthritis score of each knee joint was classified according to Kellgren/Lawrence (KL, [21]). Table 1 shows the characteristics of the study population.

The femoral condylar offset parameters and length of femoral condyles were measured with established methods. The ACO_med_, ACO_lat_, PCO_med_, PCO_lat_, LMC and LLC were determined [5,7,22]. To measure the femoral condylar offset parameters, the anterior and posterior femoral shaft bone cortex have to be determined in the sagittal view. Using measurement software, the position of each anterior and posterior femoral shaft axis was projected in the image, which best represented the anterior/posterior femoral condyle. The distance between the cortical bone of the anterior femoral condyle and the anterior cortex axis defines the ACO. According to the same principle, the PCO was defined in the posterior compartment, considering the posterior cortex axis and posterior cortical condylar bone. ACO and PCO were measured in the medial and lateral knee joint compartments. LMC and LLC were measured in axial view. Figure 1 shows the experimental process of the presented study. Figure 2 shows the measurement methodology in principle. Furthermore, the quotient of LMC/LLC was determined. All radiographic parameters in this MRI study were manually measured separately in a standardized manner by the same observer (Y.H.) under supervision of an experienced senior radiologist (A.S.A.H.). Intraobserver reliability of the measurements of all parameters was assessed for a subset of 50 subjects by blinded re-evaluation at 2 weeks after the first measurement and using the same technique. Interobserver reliability was assessed by two observers (Y.H. and M.-P.M.) independently for 50 subjects. 

### 2.5. Statistics

For side-, age- and gender-specific analyses of ACO_med_, ACO_lat_, PCO_med_, PCO_lat_, LMC, LLC and LMC/LLC, a Mann–Whitney U test was used because data were not parametric. Likewise, a Mann–Whitney U test was implemented for the comparison of patients without and with osteoarthritis for all assessed parameters. Intra- and interobserver reliabilities were evaluated using intraclass correlation coefficients (ICC). Overall, mean ± standard deviation is stated. Statistical analysis was performed with GraphPad Prism 9.00 (GraphPad Software, San Diego, CA, USA), SPSS Statistics software version 27.0 (IBM SPSS Inc., Chicago, IL, USA) and Microsoft Excel (Microsoft Office 2016, Redmond, WA, USA). Significant differences are marked with asterisks (*** *p* < 0.001, ** *p* < 0.01, * *p* < 0.05).

## 3. Results

### 3.1. Characteristics of the Study Population

In this study, 198 males (48.4%) and 211 females (51.6%) without radiological structural damage or a Kellgren/Lawrence Score < 3 were analysed. Likewise, 197 right (48.2%) and 212 left (51.8%) knees were included in the final analysis. The mean age of the study population was 52.3 years (±16.8 (age range: 21–88 years)). A total of 108 patients were classified with a Kellgren/Lawrence Score of ≥3 and were only analysed as the osteoarthritic reference group.

### 3.2. Analysis of Offset Parameters and Length of Femoral Condyles

Table 2 and Figure 3 show the descriptive analysis of femoral condylar morphology in patients without radiological structural damage or radiological osteoarthritis (*n* = 409) including mean values, standard deviation and minimal and maximal measured values for ACO_med_, ACO_lat_, PCO_med_, PCO_lat_, LMC, LLC and LMC/LLC.

#### 3.2.1. Medial Anterior Femoral Condylar Offset

In total (*n* = 409), a mean ACO_med_ of 2.82 mm (±2.54 [0.0–11.50]) was found. The mean ACO_med_ in left knees (*n* = 212) was 3.28 mm (±2.61), and 2.32 mm (±2.36) on the right joint side (*n* = 197). For the younger patients (20–50 years, *n* = 219), a mean ACO_med_ of 2.64 mm (±2.45) was detected, while the older patients (>50 years, *n* = 190) showed a mean ACO_med_ of 3.02 mm (±2.62). The analysis yielded a mean ACO_med_ of 2.48 mm (±2.44) in female knees (*n* = 211) and 3.19 mm (±2.60) in male knees.

#### 3.2.2. Lateral Anterior Femoral Condylar Offset

Overall, a mean ACO_lat_ of 6.67 mm (±2.25 [0.0–14.10]) was found. The mean ACO_lat_ in left knees was 6.61 mm (±2.36) and 6.73 mm (±2.14) in right knees. For the younger age group, a mean ACO_lat_ of 6.58 mm (±2.30) was detected, while the older patients showed a mean ACO_lat_ of 6.77 mm (±2.20). The analysis yielded a mean ACO_lat_ of 6.46 mm (±2.29) in female knees and 6.89 mm (±2.20) in male knees.

#### 3.2.3. Medial Posterior Femoral Condylar Offset

A mean PCO_med_ of 25.72 mm (±4.55 [7.30–38.10]) was found in the 409 included patients without manifest osteoarthritis. The mean PCO_med_ in left knees was 24.78 mm (±4.26), and in right knees was 26.73 mm (±4.65). For the younger age group, a mean PCO_med_ of 26.10 mm (±4.81) was detected, while the older patients showed a mean PCO_med_ of 25.27 mm (±4.20). The analysis yielded a mean PCO_med_ of 25.55 mm (±4.40) in female knees and 25.90 mm (±4.71) in male knees.

#### 3.2.4. Lateral Posterior Femoral Condylar Offset

The analysis revealed a total mean PCO_lat_ of 23.59 mm (±3.04 [3.00–32.10]). The mean PCO_lat_ in the left knees was 23.76 mm (±3.15) and on the right joint side, mean PCO_lat_ was 23.39 mm (±2.91). For the younger age group, a mean PCO_lat_ of 23.57 mm (±3.44) was detected. The older patients showed a mean PCO_lat_ of 23.62 mm (±2.51). The analysis yielded a mean PCO_lat_ of 23.09 mm (±3.24) in female knees and 24.12 mm (±2.72) in male knees.

#### 3.2.5. Length of Medial Femoral Condyle

In total, a mean LMC of 63.36 mm (±4.96 [51.40–76.40]) was found. The mean LMC in left knees was 63.27 mm (±5.15), and in right knees was 63.46 mm (±4.75). For the younger age group, a mean LMC of 63.30 mm (±5.16) was detected, while the older patients showed a mean LMC of 63.43 mm (±4.72). The analysis yielded a mean LMC of 60.28 mm (±3.64) in female knees and 66.64 mm (±3.98) in male knees.

#### 3.2.6. Length of Lateral Femoral Condyle

Overall, a mean LLC of 64.38 mm (±4.99 [52.40–88.20]) was found. The mean LLC in left knees was 64.63 mm (±5.35), and in right knees was 64.11 mm (±4.57). For the younger age group, a mean LLC of 64.36 mm (±5.03) was detected, while the older patients showed a mean LLC of 64.40 mm (±4.95). The analysis yielded a mean LLC of 61.38 mm (±3.70) in female knees and 67.57 mm (±4.13) in male knees.

#### 3.2.7. Quotient of Length of Medial to Lateral Femoral Condyle

The analysis revealed a total mean LMC/LLC of 98.50% (±3.90 [73.00–111.90]). The mean LMC/LLC in left knees was 98.00% (±3.90), and in right knees was 99.00% (±3.90). For the younger age group, a mean LMC/LLC of 98.40% (±4.10) was detected, while the older patients showed a mean LMC/LLC of 98.60% (±3.70). The analysis yielded a mean LMC/LLC of 98.30% (±3.80) in female knees and 98.70% (±4.00) in male knees.

### 3.3. Analysis of Side-Specific Differences for FCM Parameters

There was a significant difference (*p* < 0.001) for ACO_med_ and PCO_med_ between left (ACO_med_: 3.28 mm (±2.61); PCO_med_: 24.78 mm (±4.26)) and right (ACO_med_: 2.32 mm (±2.36); PCO_med_: 26.73 mm (±4.65)) knee joints. Likewise, a significantly (*p* = 0.02) higher mean LMC/LCC was found in right knee joints (right: 99.0% (±3.90); left: 98.0% (±3.90)). No side-specific differences were found for ACO_lat_ (*p* = 0.449), PCO_lat_ (*p* = 0.221), LMC (*p* = 0.716) or LCC (*p* = 0.480). All results are summarised in Table 3.

### 3.4. Analysis of Age-Specific Differences for FCM Parameters

There were no significant age-specific differences for any FCM parameters between the two age groups (20–50 years and more than 50 years): ACO_med_ (*p* = 0.130), ACO_lat_ (*p* = 0.551), PCO_med_ (*p* = 0.051), PCO_lat_ (*p* = 0.956), LMC (*p* = 0.855), LLC (*p* = 0.822) and LMC/LLC (*p* = 0.710). These results are summarised in Table 4.

### 3.5. Analysis of Gender-Specific Differences for FCM Parameters

Significant gender-specific differences were found for all parameters, with the exception of PCO_med_ (*p* = 0.301) and LMC/LLC (*p* = 0.712). The mean ACO_med_ in female patients was 2.48 mm (±2.44), and 3.19 mm (±2.60) in male patients (*p* = 0.004). Likewise, a significantly (*p* = 0.019) larger ACO_lat_ was found in male knee joints (6.89 mm (±2.20)) compared to female knee joints (6.46 mm (±2.29)). The analysis showed a significantly (*p* = 0.001) higher PCO_lat_ in male patients (24.12 mm (±2.72)). The mean PCO_lat_ in female knees was 23.09 mm (±3.24). In addition, there were higher values of mean LMC and LLC for male knee joints (LMC: 66.64 mm (±3.98); LCC: 67.57 mm (±4.13)) compared to female knee joints (LMC: 60.28 mm (±3.64); LLC: 61.38 mm (±3.70)). These results are summarised in Table 5.

### 3.6. Analysis of Osteoarthritis-Specific Differences for FCM Parameters

A total of 409 (79,1%) patients were graded as KL 0–2 and 108 (20.9%) patients were graded as KL 3–4. Only LMC/LLC showed a significant difference between these two groups (*p* = 0.015). The mean LMC/LLC in patients graded as KL 0–2 was 98.5% (±3.90), while it was 99.5% (±3.70) in patients with KL 3–4. The other FCM parameters showed no significant differences between patients without and with manifest osteoarthritis: ACO_med_ (*p* = 0.084), ACO_lat_ (*p* = 0.106), PCO_med_ (*p* = 0.626), PCO_lat_ (*p* = 0.462), LMC (*p* = 0.242) and LLC (*p* = 0.726). The results are summarised in Table 6.

### 3.7. Analysis of Intraobserver and Interobserver Reliability

ICC for intraobserver reliability ranged from 0.91 to 0.99 and interobserver reliability ranged from 0.92 to 0.99, indicating excellent reliability. Taking into account the initial measurements, the control measurements conducted by the same examiner as well those conducted by a second examiner, cumulative ICC values of 0.91 to 0.99 (ACO_med_: 0.99; ACO_lat_: 0.96; PCO_med_: 0.97; PCO_lat_: 0.91; LMC: 0.94; LLC: 0.99) resulted. These results are summarised in Figure 4.

## 4. Discussion

The morphology of the distal femur is complex [1,2,3,4]. Various studies showed that medial and lateral femoral condyle morphology differ in ACO, PCO and length [23,24,25,26]. In this context, the current literature raises the question of whether the existing standard implants for arthroplasty of the knee joint adequately consider the physiological morphology of femoral condyles [11,12,13]. Knowledge of the exact physiological FCM is the prerequisite for clarifying the question of whether side-, age- and gender-specific implants are necessary in order to be able to take into account possible morphological differences in the FCM. In the present study, MRI scans of 517 knee joints were examined to establish reference values of FCM parameters for future diagnostic, preoperative planning and therapeutic applicability.

Kuo et al. retrospectively studied 199 patients who underwent total knee arthroplasty to determine whether the femoral arthroplasty component adequately represented the bony resection height of the anterior and posterior condyles. The authors demonstrated that current standard implants do not appropriately reconstruct ACO and PCO [7].

In our analysis, significantly higher mean values for ACO_med_ were found on the left compared to the right joint side. In contrast to that, the mean values for PCO_med_ and LMC/LLC were higher on the right joint side. To the authors’ knowledge, there is currently no other study that describes a side dependence of ACO_med_, PCO_med_ and LMC/LLC in radiologically healthy knee joints. An explanation for these side-specific differences could be that 70–90% of the population use the right leg predominantly [27]. However, due to the retrospective nature of the present study and not using intraindividual analysis, this assumption can only be made hypothetically. In order to finally clarify this question, additional studies are necessary that specifically examine the ACO_med_, PCO_med_ and LMC/LLC between patients with left-leg and right-leg dominance and compare FCM intraindividually. If the assumption is confirmed in this type of study design, implants with side-specific configuration of the ACO and PCO would have to be used.

In this study, age-dependent differences for FCM parameters were not found. This result is consistent with the study of Wernecke et al. that analysed MRI scans of 287 male and 243 female patients in order to investigate the morphology of the posterior femoral condyles. Likewise, they found no correlation between patient age and PCO [28]. To the authors’ knowledge, there are no other studies examining age-specific differences in FCM in the current literature. Therefore, there is no evidence that age-specific implants are needed for physiological reconstruction of the FCM.

The analysis revealed significantly higher mean values for ACOmed, ACOlat, PCOlat, LMC and LLC in male patients compared to female patients. This raises the question of whether gender-specific implants are needed for knee arthroplasty [29]. Voleti et al. examined 100 patients with nonspecific knee pain via MRI to analyse gender-specific differences in PCO, condylar height and cartilage thickness for medial and lateral knee compartments [24]. The authors were able to detect sex-specific differences in the comparison of the absolute measured values for PCO_med_ and PCO_lat_, but not in the respective relation to condylar height. Therefore, they concluded that gender-specific implants may not be necessary. Our results also showed no significant gender differences for LMC/LCC and PCO_med_. However, Voleti et al. did not consider anterior FCM. As demonstrated by Kuo et al., the current standard implants do not adequately represent the anterior FCM in particular [7]. This is in accordance with the results of our study, as we found significant gender differences for ACO_med_ and ACO_lat_. Taking these results for ACO into account, the statement of Voleti et al. that gender-specific implants are not needed is debatable. Furthermore, the analysed cohort of Voleti et al.—with 100 patients compared to the 517 analysed knee joints in this study—is rather small. The results of Wernecke et al. are consistent with our results for PCO, pointing out a significant larger PCO_lat_ in male knee joints and no significant gender dependency of PCO_med_ [28]. Bao et al. examined CT scans of 80 patients to investigate posterior FCM [30]. The authors found a significantly larger mean PCO_lat_ and PCO_med_ in males in comparison to females. It has to be noted that for radiological imaging of osseous structures, CT is more suitable than MRI. Perhaps because of this, Bao et al. were able to demonstrate a significant gender-specific difference for PCO_med_. However, it must be pointed out that we also found slightly higher mean values for PCOmed in male knee joints compared to female joints, but without significant differences.

In summary, the ACO appears to exhibit gender specificity. In contrast to this, the published studies on PCO are inconclusive. The results, rather, speak for the necessity of gender-specific implants to adequately reconstruct the physiological FCM and thereby achieve the best possible postoperative outcome. To verify this hypothesis, future studies should investigate knee joint prostheses regarding their implant design. In general, a gender dependency of FCM should be investigated to confirm or to disprove these considerations. The results of the present study should be considered as a physiological reference for measured values of FCM.

The analysis of osteoarthritis-specific differences for FCM parameters showed only a significant difference for LMC/LLC, which pointed out a higher mean value in patients with radiological osteoarthritis (KL 3–4). The reason for this seems to be the increase in osteophytic rim attachments in higher stages of osteoarthritis. To the authors’ knowledge, no comparative study examining osteoarthritis-specific differences in FCM exists in the current literature. The assumption that the osteoarthritis-specific difference in LMC/LCC is due to osteophytic rim attachments should be considered clinically, as the osteophytic rim attachments can have an influence on the postoperative outcome after knee joint arthroplasty. Thus, Carender et al. showed that trochlear osteophytes are associated with an increased risk of failure of medial unicondylar knee arthroplasty [31]. In particular, the resection of medial osteophytes is important for optimal balancing of knee joint endoprostheses [32]. Taking into account the present results, the occurrence of osteophytes on the medial femoral condyle should, therefore, be conscientiously observed during knee joint arthroplasty, both in the preoperative diagnostics and intraoperatively.

These results illustrate the enormous importance of radiological diagnostics both for basic research for the possible optimisation of existing implant designs and for the preoperative planning of knee joint arthroplasty, especially in severely degenerated knee joints.

## 5. Limitations

A limitation of the study is that there was no comparison of the MRI scans with related CT imaging, because many of the included patients were only examined via MRI, but not via CT. A comparison between the different modalities seems useful. The present study could thereby have made an even better comparison with the results of other authors. In general, MRI is inferior to CT in terms of imaging osseous structures. The measurements performed could, therefore, not always be carried out in the best possible section plane. All analyses are based on interindividual comparisons and not on intraindividual measurements. For this reason, the reported side-specific differences in particular have to be confirmed in future prospective studies on intraindividual knee joint analysis. Another limitation of the study is the lack of biometric patient data. These could not be collected in the retrospective assessment. A dependence of FCM on, for example, body height and weight should be investigated in future studies.

## 6. Conclusions

In the present study, the MRIs of 517 patients were examined in order to analyse physiological reference values for femoral condylar morphology in a collective of radiologically healthy knee joints. Mean values for ACO_med_, ACO_lat_, PCO_med_, PCO_lat_, LMC, LLC and LMC/LLC depending on joint side, age, gender and osteoarthritis were presented. ACO_med_ showed significantly higher mean values on the left joint side, while PCO_med_ had significantly higher mean values on the right joint side. A possible reason for this is right-side dominance. In addition, significantly larger measurement results were found in males in comparison to females for all FCM parameters, with the exception of PCO_med_ and LMC/LLC. These results raise the question of whether gender-specific implants replicating physiological FCM are necessary. Further studies on this topic are needed to be able to give a substantiated answer. Age and osteoarthritis seem to have no relevant influence on the femoral condylar morphology. The presented data should be considered in knee joint diagnostics and surgery, especially in designing individual and gender-specific arthroplasty implants as well as for the preoperative planning of knee joint arthroplasty, especially in severely degenerated knee joints.

## Figures and Tables

**Figure 1 diagnostics-13-00350-f001:**
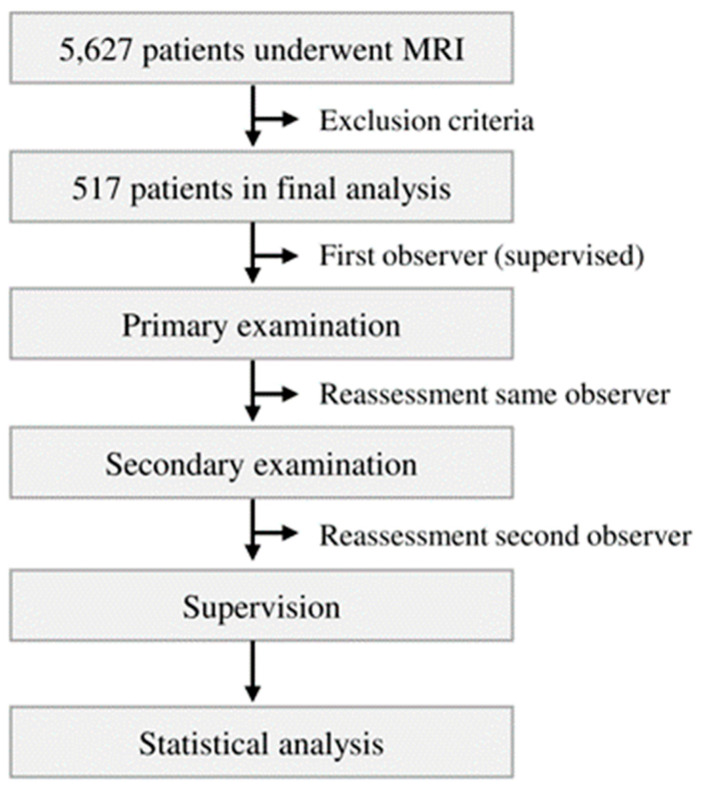
Flowchart about the experimental process of the presented study. After applying the described exclusion criteria, 517 MRIs were integrated into the final examination. Statistical analysis was performed after supervision process of examination.

**Figure 2 diagnostics-13-00350-f002:**
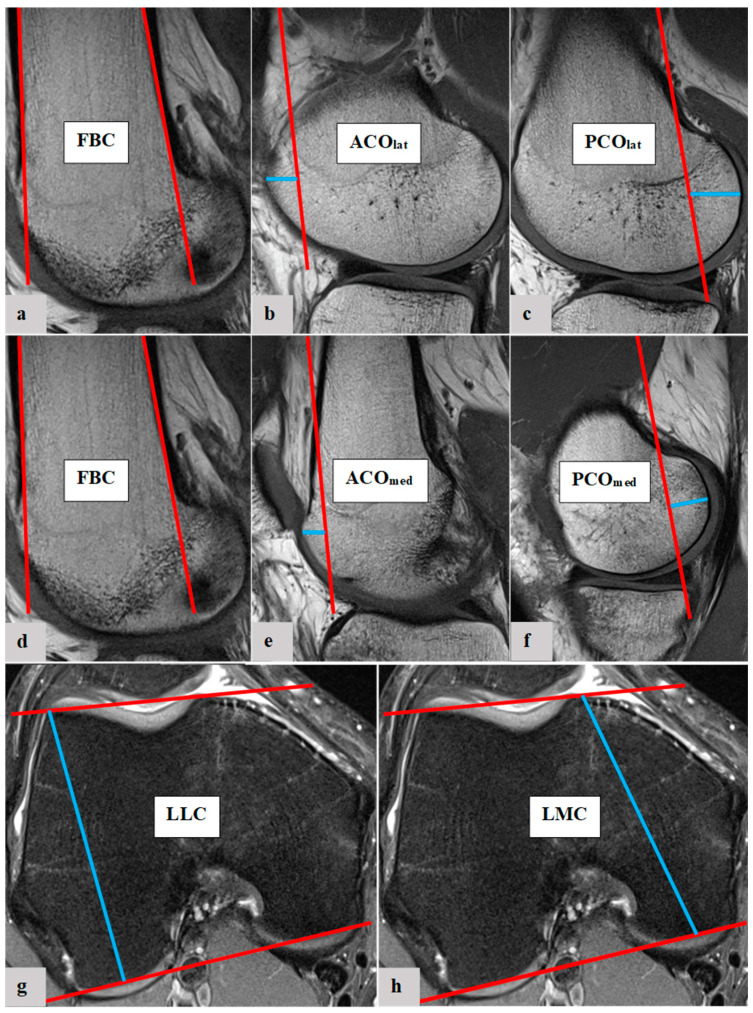
Exemplary depiction of the determination of the femoral bone cortex (FBC, (**a**,**d**)) and the measurement of lateral anterior femoral condylar offset (ACO_lat_, (**b**)), lateral posterior femoral condylar offset (PCO_lat_, (**c**)), medial anterior femoral condylar offset (ACO_med_, (**e**)), medial posterior femoral condylar offset (PCO_med_, (**f**)), length of lateral femoral condyle (LLC, (**g**)) and length of medial femoral condyle (LMC, (**h**)): The determination of FBC as well as measurements of ACO_lat_, PCO_lat_, ACO_med_ and PCO_med_ were all performed in sagittal view of the knee joint in MRIs. To measure the femoral condylar offset parameters, the anterior and posterior femoral shaft bone cortex were determined in the sagittal view (**a**,**d**). Using Centricity^TM^ Universal Viewer (RA1000, edition 2019, Buckinghamshire, Great Britain), the positions of the anterior and posterior femoral shaft axes were projected in the image, which best represented the anterior/posterior femoral condyle. The distance between the cortical bone of the anterior femoral condyle and the anterior cortex axis defines the ACO (**b**,**e**). According to the same principles, the PCO was defined in the posterior compartment. The distance between posterior cortex axis and posterior cortical condylar bone was measured (**c**,**f**). ACO and PCO were measured in the medial and lateral knee joint compartments. LLC and LMC were measured in the axial view. To define LLC and LMC, a tangent was applied to both the anterior and posterior cortical borders of the femoral condyles. The widest distance between these was measured for the lateral (**g**) and medial condyles (**h**).

**Figure 3 diagnostics-13-00350-f003:**
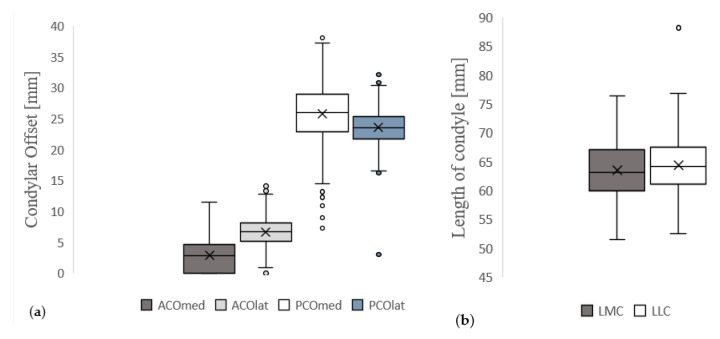
Descriptive analysis of femoral condylar morphology in patients without radiological structural damage or manifest osteoarthritis (*n* = 409): (**a**) shows the box plots of femoral condylar offset parameters (dark grey: medial anterior condylar offset (ACO_med_); light grey: lateral anterior condylar offset (ACO_lat_); white: medial posterior condylar offset (PCO_med_); blue: lateral posterior condylar offset (PCO_lat_)). (**b**) shows the box plots of the length of femoral condyles (dark grey: length of medial femoral condyle (LMC); white: length of lateral femoral condyle (LLC)).

**Figure 4 diagnostics-13-00350-f004:**
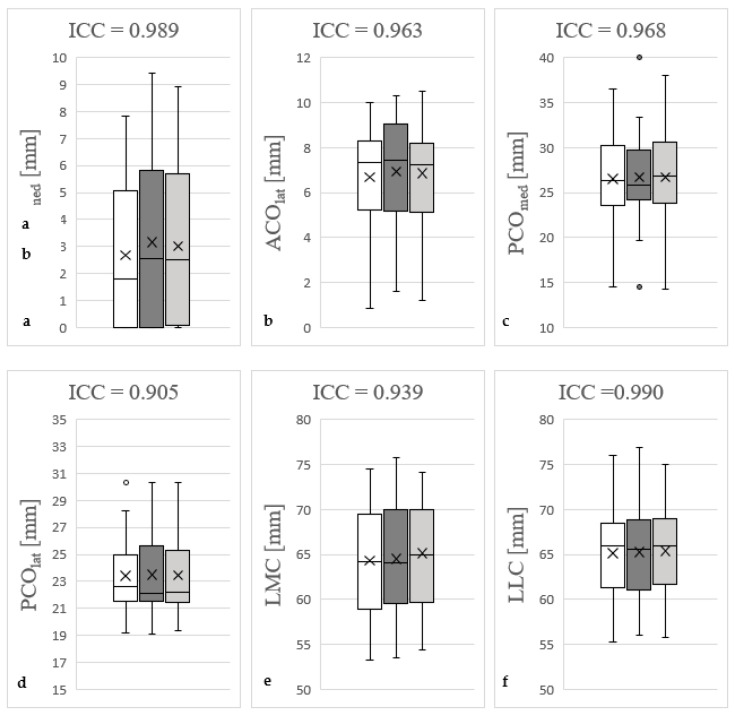
Determination of intraclass/interclass correlation coefficient (ICC) of medial anterior femoral condylar offset (ACO_med_), (**a**); lateral anterior femoral condylar offset (ACO_lat_), (**b**); medial posterior femoral condylar offset (PCO_med_), (**c**); lateral posterior femoral condylar offset (PCO_lat_), (**d**); length of medial femoral condyle (LMC), (**e**); and length of lateral femoral condyle (LLC), (**f**). The white box plots represent the initial measurements by Y.H., the dark grey ones represent the second recordings by Y.H. The box plots marked in light grey reflect the measurements by M.-P.M. The figures show the cumulative ICC between the initial and control measurements by Y.H. and the observations by M.-P.M. Outliers are marked with points. Excellent measurement reliability (ICC > 0.9) was determined for all measured parameters. The control measurements were carried out at intervals of two weeks by the same investigator as well as another one in blinded fashion based on 50 subjects.

**Table 1 diagnostics-13-00350-t001:** Characteristics of the study population (*n* = 517) divided up into patients without (KL 0–2; *n* = 409) and with osteoarthritis (KL 3–4; *n* = 108).

		KL 0–2 (*n* = 409)	KL 3–4 (*n* = 108)
Age	[years]	48.8 (±16.3)	65.6 (±11.3)
Males	[number]	198	46
Females	[number]	211	62

**Table 2 diagnostics-13-00350-t002:** Descriptive analysis of femoral condylar morphology in patients without radiological structural damage or manifest osteoarthritis (*n* = 409).

		Mean	SD	Min.	Max.
ACO_med_	[mm]	2.82	2.54	0.00	11.50
ACO_lat_	[mm]	6.67	2.25	0.00	14.10
PCO_med_	[mm]	25.72	4.55	7.30	38.10
PCO_lat_	[mm]	23.59	3.04	3.00	32.10
LMC	[mm]	63.36	4.96	51.50	76.40
LLC	[mm]	64.38	4.99	52.40	88.20
LMC/LLC	[%]	98.50	3.90	73.00	111.90

SD: standard deviation; min.: minimal; max.: maximal.

**Table 3 diagnostics-13-00350-t003:** Analysis of side-specific differences for femoral condylar parameters in patients with radiologically healthy knee joints (*n* = 409).

		Total (*n* = 409)	Left (*n* = 212)	Right (*n* = 197)	*p*-Value
ACO_med_	[mm]	2.82 (±2.54)	3.28 (±2.61)	2.32 (±2.36)	<0.001 ***^,1^
ACO_lat_	[mm]	6.67 (±2.25)	6.61 (±2.36)	6.73 (±2.14)	0.449 ^1^
PCO_med_	[mm]	25.72 (±4.55)	24.78 (±4.26)	26.73 (±4.65)	<0.001 ***^,1^
PCO_lat_	[mm]	23.59 (±3.04)	23.76 (±3.15)	23.39 (±2.91)	0.221 ^1^
LMC	[mm]	63.36 (±4.96)	63.27 (±5.15)	63.46 (±4.75)	0.716 ^1^
LLC	[mm]	64.38 (±4.99)	64.63 (±5.35)	64.11 (±4.57)	0.480 ^1^
LMC/LLC	[%]	98.50 (±3.90)	98.00 (±3.90)	99.00 (±3.90)	0.020 *^,1^

^1^ Mann–Whitney U test. (*** *p* < 0.001, * *p* < 0.05).

**Table 4 diagnostics-13-00350-t004:** Analysis of age-specific (age in years) differences for femoral condylar parameters in patients with radiologically healthy knee joints (*n* = 409).

		Total (*n* = 409)	20–50 y. (*n* = 219)	>50 y. (*n* = 190)	*p*-Value
ACO_med_	[mm]	2.82 (±2.54)	2.64 (±2.45)	3.02 (±2.62)	0.130 ^1^
ACO_lat_	[mm]	6.67 (±2.25)	6.58 (±2.30)	6.77 (±2.20)	0.551 ^1^
PCO_med_	[mm]	25.72 (±4.55)	26.10 (±4.81)	25.27 (±4.20)	0.051 ^1^
PCO_lat_	[mm]	23.59 (±3.04)	23.57 (±3.44)	23.62 (±2.51)	0.956 ^1^
LMC	[mm]	63.36 (±4.96)	63.30 (±5.16)	63.43 (±4.72)	0.855 ^1^
LLC	[mm]	64.38 (±4.99)	64.36 (±5.03)	64.40 (±4.95)	0.822 ^1^
LMC/LLC	[%]	98.50 (±3.90)	98.40 (±4.10)	98.60 (±3.70)	0.710 ^1^

^1^ Mann–Whitney U test.

**Table 5 diagnostics-13-00350-t005:** Analysis of gender-specific differences for femoral condylar parameters in patients with radiologically healthy knee joints (*n* = 409).

		Total (*n* = 409)	Females (*n* = 211)	Males (*n* = 198)	*p*-Value
ACO_med_	[mm]	2.82 (±2.54)	2.48 (±2.44)	3.19 (±2.60)	0.004 **^,1^
ACO_lat_	[mm]	6.67 (±2.25)	6.46 (±2.29)	6.89 (±2.20)	0.019 *^,1^
PCO_med_	[mm]	25.72 (±4.55)	25.55 (±4.40)	25.90 (±4.71)	0.301 ^1^
PCO_lat_	[mm]	23.59 (±3.04)	23.09 (±3.24)	24.12 (±2.72)	0.001 **^,1^
LMC	[mm]	63.36 (±4.96)	60.28 (±3.64)	66.64 (±3.98)	<0.001 ***^,1^
LLC	[mm]	64.38 (±4.99)	61.38 (±3.70)	67.57 (±4.13)	<0.001 ***^,1^
LMC/LLC	[%]	98.50 (±3.90)	98.30 (±3.80)	98.70 (±4.00)	0.172 ^1^

^1^ Mann–Whitney U test. (*** *p* < 0.001, ** *p* < 0.01, * *p* < 0.05).

**Table 6 diagnostics-13-00350-t006:** Analysis of osteoarthritis-specific differences for femoral condylar parameters between patients without (*n* = 409) and with osteoarthritis (*n* = 108).

		Total (*n* = 517)	KL 0–2 (*n* = 409)	KL 3–4 (*n* = 108)	*p*-Value
ACO_med_	[mm]	2.93 (±2.59)	2.82 (±2.54)	3.35 (±2.76)	0.084 ^1^
ACO_lat_	[mm]	6.76 (±2.31)	6.67 (±2.25)	7.09 (±2.50)	0.106 ^1^
PCO_med_	[mm]	25.69 (±4.69)	25.72 (±4.55)	25.60 (±5.19)	0.626 ^1^
PCO_lat_	[mm]	25.55 (±2.95)	23.59 (±3.04)	23.39 (±2.62)	0.462 ^1^
LMC	[mm]	63.49 (±4.57)	63.36 (±4.96)	64.00 (±4.52)	0.242 ^1^
LLC	[mm]	64.37 (±4.90)	64.38 (±4.99)	64.35 (±4.60)	0.726 ^1^
LMC/LLC	[%]	98.70 (±3.90)	98.50 (±3.90)	99.50 (±3.70)	0.015 *^,1^

^1^ Mann–Whitney U test. * *p* < 0.05.

## Data Availability

All data generated or analysed during this study are included in this published article.

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
