# Peer review of "Physiological Femoral Condylar Morphology in Adult Knees—A MRI Study of 517 Patients"

_diagnostics, 2023, doi:10.3390/diagnostics13030350_

Round 1

Reviewer 1 Report

1. A flowchart about describing the experimental process is needed.

2. In general, the characteristics of the study population should showed by a table and including BMI would be highly recommended.

3. Please add more discussion about the mechanism of osteoarthritis-specific differences for FCM.

Author Response

Reviewer 1:

Thank you very much for your feedback and constructive suggestions for improvement. They were very helpful. After revising the manuscript, we are sure that we have improved the scientific quality of the article with your support.

Point 1:

A flowchart about describing the experimental process is needed.

Response 1:

We thank the reviewer for this advice and added a flowchart about the experimental process of the presented study (s. Fig. 1; l. 114 ff.). We hope that the study process is now even easier to understand.

Point 2:

In general, the characteristics of the study population should showed by a table and including BMI would be highly recommended.

Response 2:

We thank the reviewer for this advice and added a table, which describes the characteristics of the study population (s. Tab. 1; l. 81 ff.).

We think that this addition is very useful, as it makes the division of the collective into patients without and with radiologically manifest osteoarthritis more understandable.

We fully agree with the reviewer that a consideration of biometric data would have further enhanced the quality of the present study. However, due to the retrospective study design, these could not be collected. In our opinion, the dependence of femoral condylar morphology on biometric data should be investigated in future studies.

Therefore, we added the following under the listed limitations:

“Another limitation of the study is the lack of biometric patient data. These could not be collected in the retrospective assessment. A dependence of FCM on, for example, body height and weight should be investigated in future studies.” – line 432 ff.

Point 3:

Please add more discussion about the mechanism of osteoarthritis-specific differences for FCM.

Response 3:

We thank the reviewer for this advice and complemented the discussion:

“The assumption that the osteoarthritis-specific difference in LMC/LCC is due to osteophytic rim attachments should be considered clinically, as the osteophytic rim attachments can have an influence on the postoperative outcome after knee joint arthroplasty. Thus, Carender et al. showed that trochlear osteophytes are associated with an increased risk of failure of medial unicondylar knee arthroplasty [31]. In particular, the resection of medial osteophytes is important for optimal balancing of knee joint endoprostheses [32]. Taking into account the present results, the occurrence of osteophytes on the medial femoral condyle should therefore be conscientiously observed during knee joint arthroplasty, both in the preoperative diagnostics and intraoperatively.” – line 411 ff.

Reviewer 2 Report

Overall comments for the authors:

I commend you on your submission, and thank you for your efforts.   

Line by line issues and manuscript critique:

TITLE: Does the title clearly portray the subject and purpose of the study?

The title is fine as it stands

    ABSTRACT: Does the abstract accurately reflect the study? Are all pertinent finds included?

“for all parameters significantly higher assets were measured in male knees.” - is this what you truly mean to say?

Otherwise I think the abstract is fine.

    INTRODUCTION: What is the authors' original research question, and does their study support or fulfill it?

The authors note that there has been increasing attention in total knee arthroplasty to gender and side specific differences and offset and femoral condylar morphology. Some implants, such as some of the ones used at my home institution, are actually gender specific. Thus, I think this is a particularly interesting topic.

-Lack of line numbers makes it extremely difficult to review this paper.

This is actually a really well written introduction and an excellent discussion of the importance of femoral condylar morphology and why we need to be considering it for total joint arthroplasty.

To my literature search, there are other similar studies which utilize MRI to evaluate femoral condylar morphology in normal cohorts, but I do not think that this specific study has been done previously looking specifically at condylar offset, so this is novel to my review.

    METHODS: Was the research method or study design appropriate? Is it presented sufficiently so that other researchers can duplicate them? Are the sample sizes adequate? Are the statistical analyses appropriate and correct?

The methods are described in excellent detail, and a lot of work went into this study, with senior radiologist evaluating all images. The statistical methodology appears appropriate, with Mann Whitney U being appropriate in this setting with nonparametric data here. Did you assess for whether or not the data was parametric I do not see mention of that in the statistics?

Was a power analysis performed? how do you know that you have not just obtained a very large sample size, and thus things become statistically significant that may or may not have been significant at a smaller sample size? In any case, I think a power analysis would have been appropriate.

otherwise, I think the methodologies are well described and the image is very helpful for following all the various acronyms for femoral offset and condylar measurements.

    RESULTS: Do the results answer the original research questions, as demonstrated in the Results section and tables and figures?

  Here you say the sample size is 409, which is a different number from what was reported in the title in the five hundreds. -- oh I see now, it is because you created two cohorts, one with arthritis and one without.

Other studies have previously reported gender specific differences in the morphology of the knee, (as below), so this finding is not exactly completely novel. However, I don't think any study has utilized a high number of MRI scans like they have and quantified as elegantly, so I do think that they contribute to the literature in that regard.

https://pubmed.ncbi.nlm.nih.gov/29773405/

results are well reported.

    DISCUSSION: Is the Discussion balanced? Does it put the results in context? Do the authors acknowledge the limitations of the study?

The first paragraph sums up nicely why this paper has merit, it is well known that there are differences in femoral condylar morphology, but it is certainly nice to have a large sample like this to be able to truly quantify the measurements.

“To the authors' knowledge, there is currently no other study that describes a side -dependence of ACOmed, PCOmed and LMC/LLC in radiologically healthy knee joints” - a very interesting finding, but debatably not very clinically relevant. If you take a large enough sample, almost anything becomes statistically significant.

Line 89 – “Taken these results for the ACO into account” – fix this

“the statement of Voleti et al. that gender-specific implants are not needed cannot be sustained.” - I disagree with this, show me the strong evidence that suggests that gender specific total needs do better than non gender specific....

“The results rather speak for the necessity of gender-specific implants to adequately re construct the physiological FCM and thereby achieve the best possible postoperative outcome.” - again I think this is too strong of a statement. This should be revised, to at least suggest that further evidence is needed in actual patients.

    CONCLUSIONS: Are the conclusions supported by the study findings? Does the study provide new, unique, or confirmatory findings? Will the findings be of interest to clinicians or to the public?

Conclusions are very reasonable based on what they have shown here except for this statement:

“These results suggest the need of gender -specific implants for knee joint replacement”

-       This is just too strong of a statement, the clinical literature regarding gender specific implants just isn't robust enough - they really don't seem to do any better than non-gender specific.

    TABLES AND FIGURES: Are all data presented in the text and tables and figures consistent? Do the tables clearly present information not easily summarized in the text of the paper? Are all of the tables necessary? Are the figures necessary and appropriate? Are they of high quality and clearly labeled? Can any be deleted?

tables and figures appear appropriate, very well described in the figure legends, and as I mentioned above, the figure is very helpful for determining what each of all the measurements were appeared.

    REFERENCES: Is the References section complete, or is it excessive? Does it include all of the necessary current, relevant sources

 References appear fine

Author Response

Reviewer 2:

We thank the reviewer for the positive feedback and constructive suggestions for improvment. We have modified our manuscript accordingly. In the following, we will comment on it in detail.

Point 1:

TITLE: Does the title clearly portray the subject and purpose of the study?

The title is fine as it stands

Response 1:

We thank the reviewer for the positive feedback.

Point 2:

  ABSTRACT: Does the abstract accurately reflect the study? Are all pertinent finds included?

“for all parameters significantly higher assets were measured in male knees.” - is this what you truly mean to say?

Otherwise I think the abstract is fine.

Response 2:

We apologise for the unclear phrasing. We have rewritten this section and hope that it is now more understandable.

“Except for PCOmed, the mean values of all other FCM parameters were significantly higher in male knees compared to female knees.”

Point 3:

  INTRODUCTION: What is the authors' original research question, and does their study support or fulfill it?

The authors note that there has been increasing attention in total knee arthroplasty to gender and side specific differences and offset and femoral condylar morphology. Some implants, such as some of the ones used at my home institution, are actually gender specific. Thus, I think this is a particularly interesting topic.

Response 3:

We thank the reviewer for the positive feedback. We agree with the reviewer and firmly believe that the morphology of the femoral condyles is of major importance for knee arthroplasty.     

Point 4:

-Lack of line numbers makes it extremely difficult to review this paper.

Response 4:

We apologise for this circumstance. There were line numbers in the submitted Word file, so we assume there was an error when compiling the file with the journal template. Nevertheless, we regret the difficult conditions.     

Point 5:

This is actually a really well written introduction and an excellent discussion of the importance of femoral condylar morphology and why we need to be considering it for total joint arthroplasty.

To my literature search, there are other similar studies which utilize MRI to evaluate femoral condylar morphology in normal cohorts, but I do not think that this specific study has been done previously looking specifically at condylar offset, so this is novel to my review.

Response 5:

We are very honoured and thank the reviewer for this positive feedback.

Point 6:

METHODS: Was the research method or study design appropriate? Is it presented sufficiently so that other researchers can duplicate them? Are the sample sizes adequate? Are the statistical analyses appropriate and correct?

The methods are described in excellent detail, and a lot of work went into this study, with senior radiologist evaluating all images. The statistical methodology appears appropriate, with Mann Whitney U being appropriate in this setting with nonparametric data here. Did you assess for whether or not the data was parametric I do not see mention of that in the statistics?

Was a power analysis performed? how do you know that you have not just obtained a very large sample size, and thus things become statistically significant that may or may not have been significant at a smaller sample size? In any case, I think a power analysis would have been appropriate.

otherwise, I think the methodologies are well described and the image is very helpful for following all the various acronyms for femoral offset and condylar measurements.

Response 6:

We thank the reviewer for the positive feedback and the constructive comments. In the following we would like to refer to these.

Our analysis showed that the data was not parametric. Because of that the Mann Whitney U test was implemented. We added this fact to our manuscript: “For side-, age- and gender-specific analyses of ACOmed, ACOlat, PCOmed, PCOlat, LMC, LLC and LMC/LLC, a Mann-Whitney U test was used because data was not parametric.” – l. 163 ff.

A power analysis was conducted as part of the study protocol (Cohen's d: 0.2, alpha level: 0.05, power: 0.8). This resulted in an n of 310 per group (620 in total).

5,627 MRIs were performed during the study time frame. As described in the manuscript, the majority had to be excluded from the present study due to lack of applicability. The 517 MRIs which were included in the final analysis are thus below the required number of 620 (group without manifest osteoarthritis even just 409), so that we do not assume that results became falsely significant due to the large collective size.

Point 7:

RESULTS: Do the results answer the original research questions, as demonstrated in the Results section and tables and figures?

  Here you say the sample size is 409, which is a different number from what was reported in the title in the five hundreds. -- oh I see now, it is because you created two cohorts, one with arthritis and one without.

Other studies have previously reported gender specific differences in the morphology of the knee, (as below), so this finding is not exactly completely novel. However, I don't think any study has utilized a high number of MRI scans like they have and quantified as elegantly, so I do think that they contribute to the literature in that regard.

https://pubmed.ncbi.nlm.nih.gov/29773405/

results are well reported.

Response 7:

We apologise for the difficulty in understanding the group formation. In order to improve the comprehensibility, we have created a new table (Tab. 1) which makes the division easier to understand. – 81 ff.

In addition, we thank the reviewer for the positive feedback about the reporting of our study results and the results in general.

Point 8:

DISCUSSION: Is the Discussion balanced? Does it put the results in context? Do the authors acknowledge the limitations of the study?

The first paragraph sums up nicely why this paper has merit, it is well known that there are differences in femoral condylar morphology, but it is certainly nice to have a large sample like this to be able to truly quantify the measurements.

“To the authors' knowledge, there is currently no other study that describes a side -dependence of ACOmed, PCOmed and LMC/LLC in radiologically healthy knee joints” - a very interesting finding, but debatably not very clinically relevant. If you take a large enough sample, almost anything becomes statistically significant.

Line 89 – “Taken these results for the ACO into account” – fix this

“the statement of Voleti et al. that gender-specific implants are not needed cannot be sustained.” - I disagree with this, show me the strong evidence that suggests that gender specific total needs do better than non gender specific....

“The results rather speak for the necessity of gender-specific implants to adequately re construct the physiological FCM and thereby achieve the best possible postoperative outcome.” - again I think this is too strong of a statement. This should be revised, to at least suggest that further evidence is needed in actual patients.

Response 8:

We thank the reviewer for the constructive feedback.

Regarding the clinical relevance of the side differences, we agree with the reviewer that this is questionable based on the current literature. Nevertheless, these results should not be forgotten and should be supported or refuted by further studies.

We agree with the reviewer. We cannot prove through our study that gender-specific implants perform better than non-gender-specific implants. Therefore, we have weakened the statement: “Taken these results for the ACO into account, the statement of Voleti et al. that gender-specific implants are not needed is debatable.” – l. 382 f.

Furthermore, we have revised the section as recommended:

 “To verify this hypothesis, future studies should investigate knee joint prostheses regarding to their implant design. In general, a gender-dependency of FCM should be investigated to confirm or to disprove these considerations.” – l. 400 ff.     

Point 9:

CONCLUSIONS: Are the conclusions supported by the study findings? Does the study provide new, unique, or confirmatory findings? Will the findings be of interest to clinicians or to the public?

Conclusions are very reasonable based on what they have shown here except for this statement:

“These results suggest the need of gender -specific implants for knee joint replacement”

-       This is just too strong of a statement, the clinical literature regarding gender specific implants just isn't robust enough - they really don't seem to do any better than non-gender specific.

Response 9:

Again, we thank the reviewer for the constructive feedback. We revised this statement and think that the new version is justifiable:

“These results raise the question of whether gender-specific implants replicating physiological FCM are necessary. Further studies on this topic are needed to be able to give a substantiated answer.” – l. 453 ff.      

Point 10:

TABLES AND FIGURES: Are all data presented in the text and tables and figures consistent? Do the tables clearly present information not easily summarized in the text of the paper? Are all of the tables necessary? Are the figures necessary and appropriate? Are they of high quality and clearly labeled? Can any be deleted?

tables and figures appear appropriate, very well described in the figure legends, and as I mentioned above, the figure is very helpful for determining what each of all the measurements were appeared.

Response 10:

We feel honoured and thank the reviewer for the positive feedback.     

Point 11:

REFERENCES: Is the References section complete, or is it excessive? Does it include all of the necessary current, relevant sources

References appear fine

Response 11:

We thank the reviewer for the positive feedback.